# Experimental Study on the Transport Properties of 12 Novel Deep Eutectic Solvents

**DOI:** 10.3390/polym16131946

**Published:** 2024-07-08

**Authors:** Jing Fan, Yuting Pan, Dazhi Gao, Hongwei Qu

**Affiliations:** School of Energy and Power Engineering, Northeast Electric Power University, Jilin 132012, China; crystalfan@neepu.edu.cn (J.F.); fenhongsong@neepu.edu.cn (D.G.)

**Keywords:** deep eutectic solvents, choline chloride, thermal conductivity, viscosity

## Abstract

Deep eutectic solvents (DESs) are complex substances composed of two or three components, wherein hydrogen bond donors and acceptors engage in intricate interactions within a hydrogen bond network. They have attracted extensive attention from researchers due to their easy synthesis, cost-effectiveness, broad liquid range, good stability, and for being green and non-toxic. However, studies on the physical properties of DESs are still scarce and many theories are not perfect enough, which limits the application of DESs in engineering practice. In this study, twelve DESs were synthesized by using choline chloride and betaine as HBAs, and ethylene glycol, polyethylene glycol 600, o-cresol, glycerol, and lactic acid as HBDs. The variation rules of their thermal conductivity and viscosity with temperature at atmospheric pressure were systematically investigated. The experimental results showed that the thermal conductivity of the 1:4 choline chloride/glycerol solvent was the largest at 294 K, reaching 0.2456 W·m^−1^·K^−1^, which could satisfy the demand for high efficiency heat transfer by heat-transferring workpieces. The temperature–viscosity relationship of the DESs was fitted using the Arrhenius model, and the maximum average absolute deviation was 6.77%.

## 1. Introduction

In light of the pressing global imperative for energy conservation and emission reduction, enhancing the heat transfer efficiency of heat transfer equipment has emerged as a critical concern [1,2,3,4]. Traditional heat transfer fluids are constrained by a narrow liquid range and limited thermal conductivity efficiency, rendering them inadequate for meeting the heat transfer media requirements of industrial equipment operating under complex conditions. Ionic liquids (ILs) have emerged as a promising alternative to conventional heat transfer fluids owing to their elevated boiling points, low vapor pressures, and enhanced stability. However, the drawbacks of ILs such as cumbersome synthesis steps, high cost, and poor biodegradability [5,6,7,8] limit their large-scale application.

Deep eutectic solvents are mixtures formed by the interaction of hydrogen bond donors (HBDs) and hydrogen bond acceptors (HBAs) through a large hydrogen bonding network [9,10], and are usually composed of two or three components. DESs have the advantages of a low melting point, low saturated vapor pressure, being environmentally friendly, produced from cheap raw materials, and easy to synthesize, and they have found extensive applications as environmentally sustainable solvents across diverse domains such as material extraction, electrochemical deposition, organic synthesis, and industrial catalysis [11,12,13,14,15,16].

DESs can be stabilized in liquid form over a wide range of temperatures, so they have great potential as heat transfer fluids. Abbott et al. [17,18] produced the first generation of DESs using quaternary ammonium salts, amines, or carboxylic acids as HBDs. Liu et al. [19] added nanoparticles such as TiO_2_, Al_2_O_3_ and graphene oxide to choline chloride/glycerol DESs and found that the thermal conductivity of the solvents was increased by 3–11.4%. Omar et al. [20,21] reported the melting points, densities, viscosities, electrical conductivities, and surface tensions of a variety of DESs, and illustrated the polarity, pH, toxicity, biodegradation properties, and the effect of water on DESs. Mirza et al. [22] predicted the critical properties of 39 DESs using an improved Lydersen–Joback–Reid method and Lee–Kesler mixing rules. Shaharuddin et al. [23] investigated the ionic conductivity of DESs and how ionic conductivity affects the size of pores in DES-based polymer electrolytes. However, compared with ILs, there are some limitations in the studies on the thermophysical properties of DESs, which are relatively scarce in terms of physical property data, and the underlying physical and chemical properties and mechanisms of action need to be improved.

To address the substantial gaps in the physical property data pertaining to DESs and to overcome the limitations associated with DESs in conventional heat transfer applications, this study focused on the synthesis of twelve DESs, utilizing choline chloride and betaine as HBAs and a heating and stirring methodology. Their thermal conductivity in the temperature range of 294 K to 334 K and viscosity in the temperature range of 293 K to 353 K were systematically investigated.

## 2. Experimental Section

### 2.1. Experimental Reagents

The seven chemical reagents used in this study were purchased from Shanghai Aladdin Biochemical Technology Co., Ltd. (Shanghai, China). As detailed in Table 1, and all reagents were not purified further.

### 2.2. Preparation of DESs

A PX224ZH/E electronic balance was purchased from OHAUS Instruments Co., Ltd. (Shanghai, China), with a graduation value of 0.0001 g, and was calibrated using standard weights; a DF-101SA type heat-collecting magnetic stirrer was purchased from Jiangsu Keranalytical Instrument Co., Ltd. (Changzhou, China); and a DZF-6020AB vacuum drying box was purchased from Shanghai Lichenbangxi Instrument Technology Co., Ltd. (Shanghai, China). with a vacuum degree of less than 133 Pa. Among the 7 kinds of raw materials, ChCl has strong water absorption and needs to be used after drying in a drying oven. A certain molar ratio of the HBA and HBD was weighed with an electronic balance and placed in a round bottom flask, sealed with a hermetic cap, and then heated and stirred in a magnetic stirring pot. The temperature inside the pot was generally set to 80 °C, the rotor speed was set to 600~900 r/min, The heating and stirring was performed for 2 h until the formation of clarified and transparent homogeneous solutions. The solutions were cooled down to room temperature and then put into a vacuum drying box to dry for more than twelve hours. The preparation process of the solvent is shown in Figure 1.

The twelve DESs prepared were all in the liquid state at room temperature, as shown in Table 2. They exhibited robust stability, with no discernible changes observed after being sealed and left at room temperature under light protection for a period of two months. Among them, the [ChCl][PEG600][Gl]_1:5:2_ solvent changed from a colorless transparent liquid to a light white liquid, and the choline chloride/o-cresol solvent slightly darkened the color of the solution after being placed for a period of time due to the oxidation reaction between the o-cresol and the remaining air in the bottle.

### 2.3. Measurement of Thermal Conductivity

Based on the experimental principle of the transient hot-wire theory [24,25,26,27], the experimental system for measuring transient hot-wire thermal conductivity was built. The transient hot-wire theory is recognized as one of the most reliable methods for measuring the thermal conductivity of the liquid phase, with the advantages of fast measurement speed and accurate results. The ideal model is to place an infinitely long hot wire in an infinitely large amount of fluid and apply a constant voltage to the hot wire so that the thermal conductivity of the measured liquid can be derived according to the relationship between the temperature rise and the logarithm of time. There is a thin platinum wire in the thermal conductivity measurement device that is 85 mm long and has a diameter of 20 μm, which is used as a pressure-measuring lead, which can reduce the influence of non-ideal factors. In addition, a platinum wire lead of the same thickness was welded at 10 mm from each end of the first end of the platinum wire, so that the end effects of the platinum wire can be compensated for [28,29]. The thermal conductivity experimental system is shown in Figure 2. The pressure and flow rate of the sample to be measured are controlled by a peristaltic pump; the sample is then injected into a thermal conductivity measuring device, where the temperature change in the system is precisely controlled by a thermostatic oil bath, and finally the final experimental data are integrated and calculated by a computer.

The experimental system can measure the thermal conductivity of the liquid phase in the pressure range of 0.1~20 MPa and the temperature range of 243.15~393.15 K; the temperature control accuracy is ±0.05 K, the length uncertainty of the platinum wire between the measured pressure measurement leads is 0.02%, the uncertainty of the temperature parameter of the platinum wire is in the range of 0.2%, the calorific uncertainty of the hot wire is less than 0.2%, and the uncertainty of the temperature rise slope dΔ*T_id_*/dln*t* of the hot wire is less than 0.8%, The uncertainty of the system is less than 2%. The basic principle equations are as follows:(1)$$\Delta T_{\mathrm{id}}(r_{0},t)=\frac{q}{4\pi \lambda }\ln t+\frac{q}{4\pi \lambda }\ln\left(\frac{4a}{r_{0}^{2}C}\right)$$
(2)$$\lambda (T,P)=\frac{q}{4\pi k}$$
(3)$$k=\frac{d(\Delta T_{id})}{d\left(\ln t\right)}$$
where Δ*T_id_* is the ideal temperature rise of the hot wire, *r*_0_ is the radius of the hot wire, *t* is the time of heating, *q* is the amount of heating per unit length of the wire heat source, *λ* is the coefficient of thermal conductivity, *a* is the coefficient of thermal diffusion, *C* = 1.781… is the exponential of Euler’s constant, and *k* is the slope of the line fitting the temperature rise over the logarithm of the elapsed time *t* in one ideal condition.

The thermal conductivity of deionized water in the temperature range of 293~349 K and absolute ethanol in the temperature range of 276~293 K was measured at atmospheric pressure and compared with the standard values queried in NIST. The experimental results showed that the maximum relative deviation between the measured value and the standard value of ionized water thermal conductivity under normal pressure was 1.43%, and the average absolute deviation was 0.93%. The maximum relative deviation between the measured value of the thermal conductivity value of absolute ethanol and the standard value was −1.52%, and the average absolute deviation was 0.82%.

### 2.4. Measurement of Viscosity

A Brookfield viscometer was used to measure the viscosity of the DESs. The main unit of the viscometer is an RST-CC model and the thermostatic water bath is a TC-550SD model. By controlling the rotation of a cylindrical rotor in a coaxial rotating cylinder and analyzing the shear rate of change of the rotor to obtain the corresponding complete flow curve, an evaluation of the rheological behavior of the material from the initial yield stress through relaxation, recovery, and creep can be achieved. The viscometer can measure the viscosity in the temperature range of 253.15~423.15 K with a temperature uncertainty of ±0.04 K.

## 3. Experimental Results and Analysis

### 3.1. Results and Analysis of Thermal Conductivity of DESs

Thermal conductivity stands as a fundamental thermophysical property of fluids and is important for the design and optimization of heat transfer devices. In this experiment, the thermal conductivity of the twelve DESs was measured in the temperature range of 294~334 K at atmospheric pressure using the transient hot-wire method. One temperature measurement was taken every 10 K, and a total of five temperature measurements were taken for each solvent for the experiment, and the measurements were repeated four times and then averaged, and the experimental data are shown in Table 3. A quadratic polynomial was utilized for the fitting, and the fitting parameters are shown in Table 4, and the maximum deviation of the experimental values from the fitted values was 0.14%; the fitting equations are as follows:(4)$$\lambda =\mathrm{AT}+{\mathrm{BT}}^{2}+C$$

Figure 3a demonstrates the variation in thermal conductivity with temperature for the DESs. It can be seen that ChCl/Gl exhibited high thermal conductivity. Among them, the ChCl/O-cresol solvent had the smallest thermal conductivity and the thermal conductivity decreased with increasing temperature, which was opposite to the trend of the other ten solvents. The thermal conductivity of the [ChCl][Gl]_4_ solvent reached 0.2456 W·m^−1^·K^−1^ at 294 K. This is due to the fact that the magnitude of the hydrogen bonding strength influences the thermal conductivity of the solvents [30,31], and the greater electronegativity between the atoms in the glycerol can easily form a strong hydrogen-bonding network with ChCl, thus exhibiting a high thermal conductivity. Figure 3b shows the variation in thermal conductivity with temperature for [ChCl][Gl]_4_, Gl, LA, EG, and O-cresol, from which it can be seen that they can be ranked as Gl > EG > [ChCl][Gl]_4_ > LA > O-cresol. Figure 4 shows the deviation between the experiment data and the calculated data.

### 3.2. Results and Analysis of Viscosity of DESs

The viscosity of a fluid serves as a critical parameter for assessing its flow characteristics and can directly impact the necessary pumping power for the fluid [32]. Viscosity is also an important parameter for studying the properties and interactions of materials. In this experiment, the viscosity of the twelve DESs was measured in the temperature range of 293~353 K, and one temperature measurement was taken every 10 K, and a total of seven temperature measurement were taken for each solvent, and the experimental data are shown in Table 5. The experimental results show that the viscosity of the DESs was much larger than that of general organic solvents. At low temperatures, the fluid viscosity decreased significantly with increasing temperature. This can be explained at the molecular level by the fact that the increase in temperature intensifies the thermal motion of the molecules and the intermolecular interaction forces are weakened, thus making the relative motion between the fluid molecules easier.

Figure 5 shows the change in solvent viscosity with temperature. It can be seen that the solvent viscosity of ChCl/O-cresol was significantly smaller than that of the other ten solvents. At 292.54 K, the viscosity of [ChCl][O-cresol]_5_ was only 48.3 mPa·s. Compared with the ChCl/LA solvent, the viscosity increased significantly when the HBA was changed to betaine. At 293 K, the viscosity of [Bet][LA]_3_ was elevated by 360.37% compared to the viscosity of [ChCl][Gl]_3_. Therefore, suitable HBAs, HBDs, and ratios should be selected for practical applications with low viscosity requirements.

In this study, the relationship between viscosity and temperature was fitted using the Arrhenius model [33,34] with the following equation:(5)$$\ln\eta =\ln\eta_{0}+\frac{E_{\eta }}{RT}$$
where *η* is the viscosity, *η*_0_ is a constant, *E_η_* is the activation energy, *R* is the gas constant, and *T* is the temperature. Figure 6 shows the linear relationship between *T*^−1^ and ln*η*, and the *η*_0_ and activation energy *E_η_* values are shown in Table 6.

After being fitted using the Arrhenius formula, [ChCl][Gl]_3_, [ChCl][Gl]_4_, [ChCl][LA]_3_, [ChCl][LA]_5_, [ChCl][Gl][EG]_1:1:2_, [ChCl][Gl][EG]_1:2:2_, [ChCl][PEG600][Gl]_1:4:2_, [ChCl][PEG600][Gl]_1:5:2_, [ChCl][O-cresol]_4_, [ChCl][O-cresol]_5_, [Bet][LA]_3_, and [Bet][LA]_4_ had mean absolute deviations of 3.61%, 4.53%, 5.55%, 5.3%, 2.83%, 3.04%, 3.62%, 4.07%, 3.84%, 6.77%, 3.07% and 4.11%. It can be seen that there was some deviation between the experimental and fitted values of viscosity, which is due to the fact that the Arrhenius model is more suitable to be used when the fluid temperature varies less, whereas in this experiment, the temperature measurement range was larger.

## 4. Conclusions

In this study, twelve DESs were synthesized by a heating and stirring method using choline chloride and betaine as HBAs and glycerol, ethylene glycol, polyethylene glycol 600, o-cresol, and lactic acid as HBDs. The thermal conductivity and viscosity of the DESs were measured, which provided data support for the prediction of their physical properties, the construction of physical property models, and facilitating the engineering application of DESs, and laid a foundation for further research on their basic physicochemical properties and action mechanisms. The following conclusions were reached:(1)The twelve DESs appeared as homogeneous and stable liquids at room temperature, and no crystallization or precipitation was observed after 2 months of storage in a closed environment protected from light. Among them, the light yellow ChCl/O-cresol solvent slightly deepened in color, and the [ChCl][PEG600][Gl]_1:5:2_ solvent turned into a light white liquid.(2)The thermal conductivity of the DESs was measured using the experimental principle of the transient hot-wire method. The results showed that the thermal conductivity of the ChCl/O-cresol solvent decreased with increasing temperature, which was the opposite trend of the other ten solvents. The thermal conductivity of the [ChCl][Gl]_4_ solvent reached 0.2456 W·m^−1^·K^−1^ at 294 K. The thermal conductivity of the [ChCl][Gl]_4_ solvent was found to decrease with increasing temperature.(3)At low temperatures, the viscosities of the twelve solvents decreased significantly with increasing temperature. The selection of the HBA, HBD, or different ratios can have a dramatic effect on the viscosity of the solvents. The viscosity of [Bet][LA]_3_ was enhanced by 360.37% compared with that of [ChCl][Gl]_3_ at 293 K. At 293 K, the viscosity of [ChCl][O-cresol]_5_ was only 48.3 mPa·s. The activation energies of the twelve solvents were small, and all of them were in the interval of 29.36~49.31 KJ·mol^−1^.

In recent years, major breakthroughs have been made in the research on DESs. However, the current systematic research and physical property data on the physical properties of DESs are not comprehensive enough. In addition, at the molecular level, there is no unified understanding of the mechanism of DESs. In order to solve these problems, future research work should also deeply explore the internal structure and mechanism of DESs at the microscopic molecular level, reveal the interaction between molecules, and develop a theoretical model with an excellent physical property prediction ability.

## Figures and Tables

**Figure 1 polymers-16-01946-f001:**
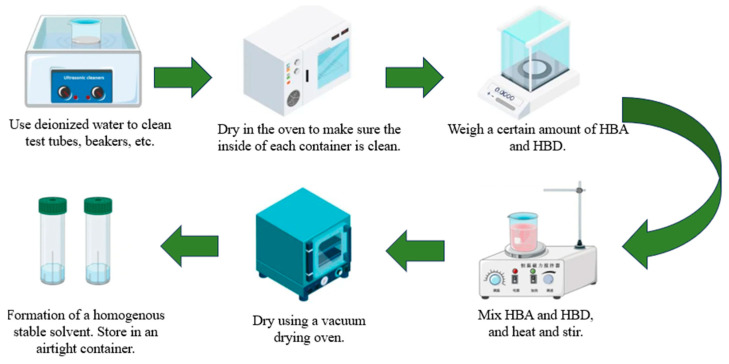
The preparation process of DESs.

**Figure 2 polymers-16-01946-f002:**
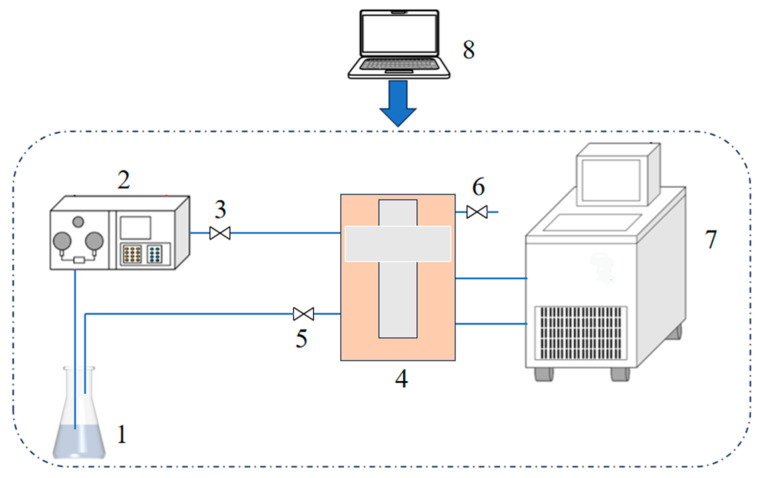
The thermal conductivity experimental system. 1. Sample bottle. 2. Peristaltic pump. 3. Injection valve. 4. Thermal conductivity measurement device. 5. Drain valve. 6. Vacuum valve. 7. Thermostatic bath. 8. Computer.

**Figure 3 polymers-16-01946-f003:**
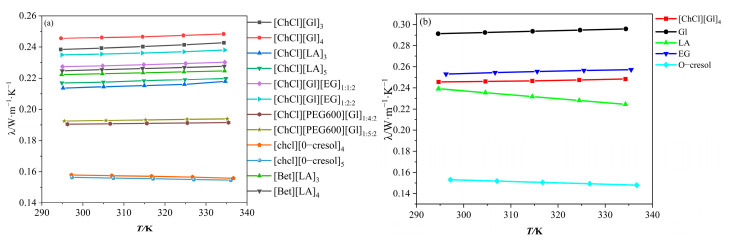
(**a**) Variation in thermal conductivity of DESs with temperature; (**b**) variation in thermal conductivity of [ChCl][Gl]_4_, Gl, LA, EG, and O-cresol with temperature.

**Figure 4 polymers-16-01946-f004:**
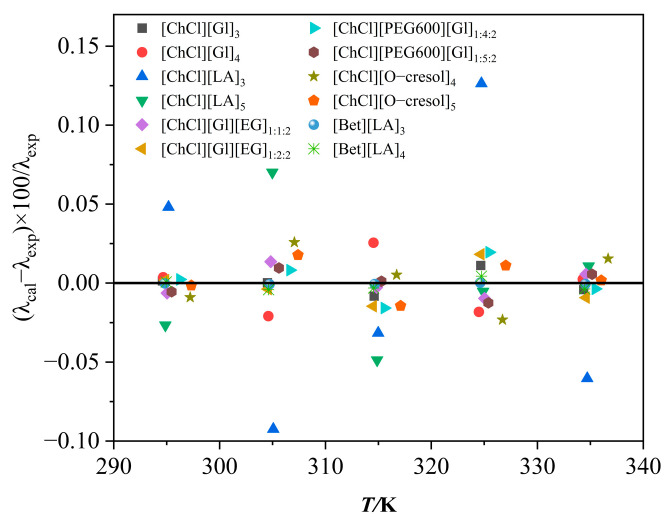
Thermal conductivity deviation of DESs.

**Figure 5 polymers-16-01946-f005:**
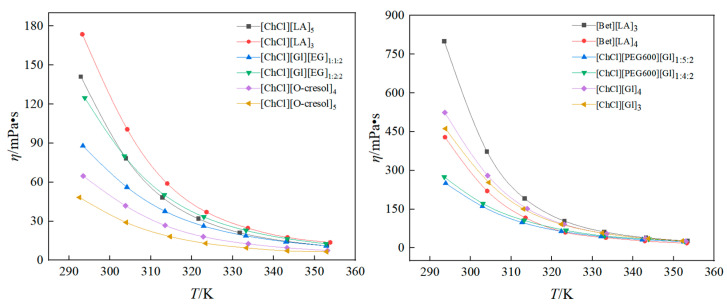
DES viscosity variation with temperature.

**Figure 6 polymers-16-01946-f006:**
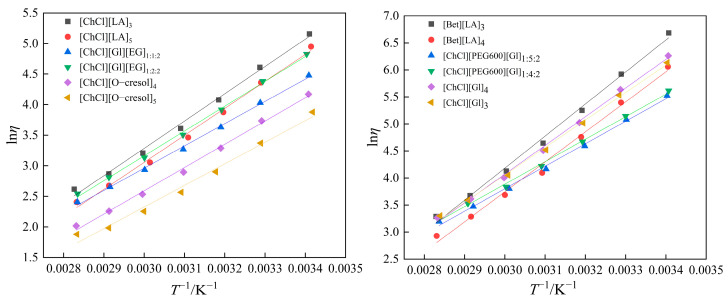
Linear relationship between *T*^−1^ and ln*η*.

**Table 1 polymers-16-01946-t001:** Experimental reagents.

Reagent Name	CAS Number	Molecular Formula	Molecular Weight (Supplier)	Purity
Choline chloride	67-48-1	C_5_H_14_ClNO	139.62	98% (AR)
Betaine	107-43-7	C_5_H_11_NO_2_	117.15	98% (AR)
Glycerol	56-81-5	C_3_H_8_O_3_	92.09	99% (AR)
Ethylene glycol	107-21-1	C_2_H_6_O_2_	62.07	98% (AR)
PEG600	25322-68-3	HO(CH_2_CH_2_O)_n_H	600	
O-cresol	95-48-7	C_7_H_8_O	108.14	98% (AR)
Lactic acid	50-21-5	C_3_H_6_O_3_	90.08	85~90% (AR)

**Table 2 polymers-16-01946-t002:** Abbreviations and characterization of DESs.

HBA	HBD	Molar Ratio		Abbreviation	State (293 K)
		HBA	HBD		
Choline chloride	Glycerol	1	3	[ChCl][Gl]_3_	Clear liquid
		1	4	[ChCl][Gl]_4_	
Choline chloride	Glycerol/Ethylene glycol	1	1:2	[ChCl][Gl][EG]_1:1:2_	Clear liquid
		1	2:2	[ChCl][Gl][EG]_1:2:2_	
Choline chloride	PEG600/Glycerol	1	4:2	[ChCl][PEG600][Gl]_1:4:2_	Clear liquid
		1	5:2	[ChCl][PEG600][Gl]_1:5:2_	
Choline chloride	O-cresol	1	4	[ChCl][O-cresol]_4_	Light brown liquid
		1	5	[ChCl][O-cresol]_5_	
Choline chloride	Lactic acid	1	3	[ChCl][LA]_3_	Clear liquid
		1	5	[ChCl][LA]_5_	
Betaine	Lactic acid	1	3	[Bet][LA]_3_	Clear liquid
		1	4	[Bet][LA]_4_	

**Table 3 polymers-16-01946-t003:** Experimental data of thermal conductivity of twelve DESs.

*T*/	*λ*/	*T*/	*λ*/	*T*/	*λ*/
K	W·m^−1^·K^−1^	K	W·m^−1^·K^−1^	K	W·m^−1^·K^−1^
[ChCl][Gl]_3_		[ChCl][Gl]_4_		[ChCl][LA]_3_	
294.59	0.2385	294.67	0.2456	295.17	0.2137
304.51	0.2393	304.61	0.2461	305.07	0.2145
314.60	0.2403	314.54	0.2466	314.98	0.2152
324.67	0.2414	324.49	0.2475	324.74	0.2160
334.36	0.2427	334.28	0.2484	334.70	0.2179
[ChCl][LA]_5_		[ChCl][Gl][EG]_1:1:2_		[ChCl][Gl][EG]_1:2:2_	
294.88	0.2169	295.04	0.2274	294.89	0.2350
304.98	0.2174	304.83	0.2279	304.58	0.2355
314.88	0.2184	314.84	0.2286	314.50	0.2362
324.89	0.2191	325.03	0.2294	324.68	0.2370
334.84	0.2199	334.58	0.2302	334.59	0.2381
[ChCl][PEG600][Gl]_1:4:2_		[ChCl][PEG600][Gl]_1:5:2_		[ChCl][O-cresol]_4_	
296.25	0.1904	295.47	0.1925	297.22	0.1579
306.65	0.1907	305.61	0.1929	307.06	0.1575
315.58	0.1910	315.29	0.1933	316.73	0.1571
325.47	0.1912	325.39	0.1937	326.73	0.1566
335.48	0.1915	335.16	0.1940	336.67	0.1559
[ChCl][O-cresol]_5_		[Bet][LA]_3_		[Bet][LA]_4_	
297.33	0.1565	294.82	0.2223	294.95	0.2248
307.41	0.1560	304.72	0.2229	304.63	0.2255
317.10	0.1556	314.65	0.2235	314.55	0.2262
327.04	0.1551	324.62	0.2241	324.73	0.2269
336.02	0.1547	334.53	0.2247	334.46	0.2276

**Table 4 polymers-16-01946-t004:** Fitting parameters for thermal conductivity.

DES	*A*	*B*	*C*	r^2^
[ChCl][Gl]_3_	−4.19084 × 10^−4^	8.33701 × 10^−7^	0.28961	0.999
[ChCl][Gl]_4_	−4.89096 × 10^−4^	8.89908 × 10^−7^	0.31246	0.999
[ChCl][LA]_3_	−9.60836 × 10^−4^	1.68473 × 10^−6^	0.35063	0.986
[ChCl][LA]_5_	−6.61081 × 10^−5^	2.27432 × 10^−7^	0.21656	0.994
[ChCl][Gl][EG]_1:1:2_	−2.63097 × 10^−4^	5.31426 × 10^−7^	0.25875	0.999
[ChCl][Gl][EG]_1:2:2_	−4.93418 × 10^−4^	9. 93418 × 10^−7^	0.30165	0.999
[ChCl][PEG600][Gl]_1:4:2_	7.51905 × 10^−5^	−7.50958 × 10^−8^	0.17472	0.997
[ChCl][PEG600][Gl]_1:5:2_	1.23432 × 10^−4^	−1.34945 × 10^−7^	0.1678	0.999
[ChCl][O-cresol]_4_	2.64003 × 10^−4^	−4.94921 × 10^−7^	0.12314	0.998
[ChCl][O-cresol]_5_	−5.1071 × 10^−5^	7.38161 × 10^−9^	0.17103	0.999
[Bet][LA]_3_	6.20657 × 10^−5^	−2.62959 × 10^−9^	0.20423	1
[Bet][LA]_4_	8.24401 × 10^−5^	−1.8784 × 10^−8^	0.20212	0.999

**Table 5 polymers-16-01946-t005:** Experimental data of viscosity of twelve DESs.

*T*/	*η*/	*T*/	*η*/	*T*/	*η*/
K	mPa·s	K	mPa·s	K	mPa·s
[ChCl][Gl]_3_		[ChCl][Gl]_4_		[ChCl][LA]_3_	
293.80	461.5	293.62	524.1	293.25	173.6
304.50	253.3	304.22	279.8	304.22	100.6
313.04	150.9	313.95	152.1	313.98	59.0
322.34	91.7	323.07	90.9	323.59	37.0
332.43	57.6	333.54	54.9	333.71	24.7
343.83	36.3	343.03	37.1	343.39	17.6
352.19	27.1	353.04	26.2	353.83	13.7
[ChCl][LA]_5_		[ChCl][Gl][EG]_1:1:2_		[ChCl][Gl][EG]_1:2:2_	
292.90	141.0	293.37	87.9	293.82	124.8
303.95	78.3	304.13	56.2	303.54	80.0
312.84	48.2	313.44	37.7	313.28	50.2
321.63	31.9	322.89	26.3	322.95	33.3
331.75	21.2	333.20	18.8	333.25	22.9
343.33	14.5	343.02	14.2	343.35	16.6
353.07	11.1	352.84	11.1	352.78	12.7
[ChCl][PEG600][Gl]_1:4:2_		[ChCl][PEG600][Gl]_1:5:2_		[ChCl][O-cresol]_4_	
293.52	274.3	293.86	250.1	293.43	64.8
303.04	171.5	302.88	160.9	303.80	41.9
313.20	106.8	312.66	98.6	313.46	26.8
323.51	68.0	322.34	64.4	322.81	18.1
333.02	46.2	332.15	44.7	333.80	12.6
343.85	34.1	342.29	32.2	343.30	9.6
352.79	25.8	352.51	24.4	353.20	7.5
[ChCl][O-cresol]_5_		[Bet][LA]_3_		[Bet][LA]_4_	
292.54	48.3	293.53	799.2	293.68	428.4
303.98	29.1	304.05	372.5	304.09	220.3
314.68	18.2	313.33	190.8	313.52	116.6
323.41	13.0	323.12	104	323.35	60.2
333.38	9.5	332.93	62.1	333.31	39.9
343.34	7.3	343.23	39.4	342.92	26.7
352.95	6.5	353.49	26.8	353.27	18.7

**Table 6 polymers-16-01946-t006:** Parameters for viscosity fitting.

DES	*η*_0_/(mPa·s)	*E_η_*/(kJ·mol^−1^)	r^2^
[ChCl][Gl]_3_	1.3781 × 10^−5^	42.26	0.998
[ChCl][Gl]_4_	7.0270 × 10^−7^	44.16	0.997
[ChCl][LA]_3_	4.0445 × 10^−5^	37.14	0.995
[ChCl][LA]_5_	3.7587 × 10^−5^	36.70	0.995
[ChCl][Gl][EG]_1:1:2_	3.4969 × 10^−4^	30.26	0.998
[ChCl][Gl][EG]_1:2:2_	1.2114 × 10^−4^	33.76	0.998
[ChCl][PEG600][Gl]_1:4:2_	1.8643 × 10^−4^	34.57	0.997
[ChCl][PEG600][Gl]_1:5:2_	1.6930 × 10^−4^	34.60	0.997
[ChCl][O-cresol]_4_	1.4793 × 10^−4^	31.62	0.996
[ChCl][O-cresol]_5_	2.5616 × 10^−4^	29.36	0.987
[Bet][LA]_3_	1.2290 × 10^−6^	49.31	0.995
[Bet][LA]_4_	2.5023 × 10^−6^	46.12	0.993

## Data Availability

Data are contained within the article.

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
