# Peer review of "Experimental Study on the Transport Properties of 12 Novel Deep Eutectic Solvents"

_polymers, 2024, doi:10.3390/polym16131946_

Round 1

Reviewer 1 Report

Comments and Suggestions for Authors

1. The uncertainty analysis should be presented.

2. How is the thermal conductivity experimental system calibrated? Calibration report should be presented.

3. Please present the graphs of shear stress versus shear rate to show that the suspensions are Newtonian.

4. To strengthen the literature review citing to the following papers are presented

Molecular dynamics simulation concerning nanofluid boiling phenomenon affected by the external electric field: Effects of number of nanoparticles through Pt, Fe, and Au microchannels

Ionic Conductivity, Dielectric, and Structural Insights of Deep Eutectic Solvent-based Polymer Electrolyte: A Review‏

Author Response

  1. The uncertainty analysis should be presented.

Response: Thank you for your valuable comments. This revision added the uncertainty analysis of the thermal conductivity measurement.

  1. How is the thermal conductivity experimental system calibrated? Calibration report should be presented.

Response: Thank you for your valuable comments. We have added the results of the thermal conductivity test system with deionised water and anhydrous ethanol.

  1. Please present the graphs of shear stress versus shear rate to show that the suspensionsare Newtonian.

Response: Thank you for your valuable comments. We apologize that the viscosity was measured by a Brookfield viscometer, and it could not show the relationship between shear stress and shear rate.

  1. To strengthen the literature review citing to the following papers are presented.

Molecular dynamics simulation concerning nanofluid boiling phenomenon affected by the external electric field: Effects of number of nanoparticles through Pt, Fe, and Au microchannels.

Ionic Conductivity, Dielectric, and Structural Insights of Deep Eutectic Solvent-based Polymer Electrolyte: A Review.

Response: Thank you for your suggestion. Two important literature have been cited in the revised version.

Reviewer 2 Report

Comments and Suggestions for Authors

General comment: This study focuses on the measurement of thermal conductivity and viscosity of twelve different types of DES. The results are indeed useful for the selection of the proper DES according to the utilization scope. However, the paper must be strengthened prior to publication. Below is a list of detailed comments:

Punctual comments:

Point 1: The introduction section is mainly based on thermal argumentation. However, there is a wide body of literature (as the authors pointed out) that deals with DESs as an alternative and "green" solvent for the extraction of bioactive molecules from solid matrices. The main limitation of DESs for industrial breakthroughs in this application is their very high viscosity. Please strengthen the introduction by emphasizing the importance of your work as a useful guide for DES selection in extraction applications based on their viscosity.

Point 2: In the experimental section, please specify if additional steps for the DES preparation are required. For example, choline chloride (CAS number 67-48-1) in powder form, due to its hygroscopic nature, is usually treated in an oven before being used for DES synthesis.

Point 3: In the experimental section, Figure 2 is just partially described. Please provide additional information in the text.

Point 4: Some comments on Figure 3: First of all, please add a letter to each plot to distinguish them (e.g., a, b, c, d). I assume that the plot of ChCl/O-Cresol was separated from the others due to the different trend exhibited by this DES with respect to temperature. However, the axis scale is very different from the two plots above, and while the decrease seems significant, the conductivity changes only slightly (this observation also applies to the plot of Betaine/LA). For this reason, I suggest including all the curves in one plot and then providing additional plots as a “zoom” of the first one. Please check also on the quality of the Figure.

Point 5: In line 142, the trend of the curve for the ChCl/O-cresol solvent was explained, but how significant is this decrease? Is it substantial? This observation is important, and I think it should be mentioned in the conclusions to provide a practical consideration.

Point 6: Figure 4 was not referenced in the text. Please reference the figure and provide a description.

Point 7: Please provide some comparisons between the conductivities of the different DES under analysis and other common fluids (e.g. water)

Point 8: The conclusions must be improved. In particular, add the implications that this study could have on DES applications and provide possible next steps for this research line.

Author Response

Point 1: The introduction section is mainly based on thermal argumentation. However, there is a wide body of literature (as the authors pointed out) that deals with DESs as an alternative and "green" solvent for the extraction of bioactive molecules from solid matrices. The main limitation of DESs for industrial breakthroughs in this application is their very high viscosity. Please strengthen the introduction by emphasizing the importance of your work as a useful guide for DES selection in extraction applications based on their viscosity.

Response: Thank you for your useful suggestion. In this work, the main purpose is to develop and design different DESs as heat transfer fluids. So review about other applications haven’t been described enough.

Point 2: In the experimental section, please specify if additional steps for the DES preparation are required. For example, choline chloride (CAS number 67-48-1) in powder form, due to its hygroscopic nature, is usually treated in an oven before being used for DES synthesis.

Response: Thank you for your professional suggestion. This paper adds the details about synthesis.

Point 3: In the experimental section, Figure 2 is just partially described. Please provide additional information in the text.

Response: Thank you for your valuable comments. This paper adds a description of Figure 2.

Point 4: Some comments on Figure 3: First of all, please add a letter to each plot to distinguish them (e.g., a, b, c, d). I assume that the plot of ChCl/O-Cresol was separated from the others due to the different trend exhibited by this DES with respect to temperature. However, the axis scale is very different from the two plots above, and while the decrease seems significant, the conductivity changes only slightly (this observation also applies to the plot of Betaine/LA). For this reason, I suggest including all the curves in one plot and then providing additional plots as a “zoom” of the first one. Please check also on the quality of the Figure.

Response: Thank you for your valuable comments. We have consolidated the four figures in the original Figure 3 into one figure, which is now Figure 3(a).

Point 5: In line 142, the trend of the curve for the ChCl/O-cresol solvent was explained, but how significant is this decrease? Is it substantial? This observation is important, and I think it should be mentioned in the conclusions to provide a practical consideration.

Response: Thank you for your valuable input. The paper concludes with the addition of observations on the trend of the curve.

Point 6: Figure 4 was not referenced in the text. Please reference the figure and provide a description.

Response: Thank you for your valuable comments. We have cited Figure 4 and described it in the text.

Point 7: Please provide some comparisons between the conductivities of the different DES under analysis and other common fluids (e.g. water).

Response: Thank you for your valuable comments. We have added the variation of thermal conductivity with temperature for [ChCl][Gl]4 with Gl, LA, EG and O-cresols as shown in Fig. 3(b) and compared the relationships between them.

Point 8: The conclusions must be improved. In particular, add the implications that this study could have on DES applications and provide possible next steps for this research line.

Response: Thank you for your valuable comments. We have added research implications and prospects in our conclusion.

Round 2

Reviewer 1 Report

Comments and Suggestions for Authors

Accept

Author Response

  1. Accept after minor revision (corrections to minor methodological errors and text editing).

Response: Thank you for your valuable comments. Based on your comments we have made the following changes:

  • Correct the name of the author of the document cited in line 45.
  • Corrected formatting and font of table
  • Format all formulas in the text and use the correct fonts.
  • Amend line 126 by indenting the first line by 0.75 cm.
  • Adjust diffusion coefficient a to italics in line 123.
  • Correction of subheading number 2.4 in line 134.
  • Corrected font and formatting of table 3.
  • Change ηand E to italics in line 196,197 and 198.
  • Add space after all heading numbers.